

# Caffeic acid phenethyl ester inhibits multispecies biofilm formation and cariogenicity

Paopanga Kokilakanit[1], Nonthakorn Dungkhuntod[1], Nitchadakorn Serikul[1], Sittichai Koontongkaew[2] and Kusumawadee Utispan[1]

[1] Faculty of Dentistry, Thammasat University, Klong Luang, Pathum Thani, Thailand
[2] International College of Dentistry, Walailak University, Dusit, Bangkok, Thailand

## ABSTRACT

**Background:** Caffeic acid phenethyl ester (CAPE), a natural phenolic compound, has demonstrated antibacterial effects. Dental caries etiology is multifactorial, including a cariogenic biofilm containing multispecies bacteria. However, the antibacterial property of CAPE on multispecies biofilm is unclear. The aim of this study was to assess the effect of CAPE on the formation and cariogenicity in biofilm containing *Streptococcus mutans*, *Streptococcus oralis*, and *Streptococcus mitis*.
**Methods:** *S. mutans* (ATCC 25175), *S. oralis* (ATCC 35037), and *S. mitis* (ATCC 49456T) were employed in this investigation. Each bacterial strain was cultured in the presence of CAPE, followed by susceptibility assessment through optical density measurements at a 600 nm wavelength. Multispecies biofilm formation was achieved by co-culturing *S. mutans*, *S. oralis*, and *S. mitis* at a 1:1:1 ratio on hydroxyapatite-coated 96-well plates. The anti-adherence activity of CAPE on multispecies biofilm was evaluated using a crystal violet staining assay. Cariogenic gene expression level and glucosyltransferase (GTF) function in CAPE-treated mixed bacteria were evaluated using real-time PCR and enzyme activity assay, respectively. The thickness and bacterial viability in CAPE-treated multispecies biofilm were examined using confocal laser scanning microscopy.
**Results:** CAPE demonstrated a significant antimicrobial effect on *S. mutans*, *S. oralis*, and *S. mitis* ($p < 0.05$). The inhibition concentration 50% (IC50) of CAPE against *S. mutans*, *S. oralis*, and *S. mitis* ranged from 1.6–6.4 mg/ml. CAPE significantly hindered the multispecies biofilm adherence ($p < 0.05$). Furthermore, the expression of genes involved in acidogenicity, aciduricity, sucrose-dependent adhesion and quorum sensing mechanism and GTF activity were significantly decreased in CAPE-treated mixed bacteria ($p < 0.05$). In a multispecies biofilm, CAPE significantly reduced its thickness and viable bacteria population ($p < 0.05$). In conclusion, CAPE exhibited antimicrobial, anti-adherence and anti-cariogenic effects within a multispecies biofilm. These findings suggest the potential use of CAPE as an adjunctive anti-cariogenic agent in future dental applications.

Corresponding author
Kusumawadee Utispan,
kusumawadee.utispan@gmail.com

## INTRODUCTION

Dental caries is a biofilm-mediated, diet modulated, non-communicable, multifactorial dynamic disease, which impacts the net mineral loss of the tooth (*Machiulskiene et al., 2020*). Dental biofilm is composed of multispecies commensal bacteria that forms on the enamel surface. The biofilm environment changes when acid increases due to sugar fermentation in bacteria (*Marsh, 1991*). The normal flora in dental biofilm ferments the sugar and generates acid as a by-product. The acidic biofilm promotes the growth of cariogenic bacteria, especially *Streptococcus mutans*, *Lactobacillus spp.*, and low pH non-mutans streptococci, such as *S. mitis* and *S. oralis* (*Marsh, 1991*; *Takahashi & Nyvad, 2008*). The ecological change in dental biofilm increases enamel demineralization (*Marsh, 1991*).

The cariogenic bacteria generate dental caries due to their acidogenicity, aciduricity, sucrose-dependent adhesion activity and quorum sensing-based adaptation mechanism (*You, 2019*). In an acidogenic biofilm, lactic acid is a product of lactate dehydrogenase (LDH) activity in the bacterial fermentation process (*Duguid, 1985*). The cariogenic bacteria contain an acid tolerance mechanism in response to a low pH environment (aciduricity). Up-regulation of *brpA* gene expression in *S. mutans* maintains membrane integrity and tolerance to acidic conditions (*Lemos & Burne, 2008*; *Wen, Baker & Burne, 2006*). *S. mutans* utilizes glucosyltransferases to catalyze sucrose into water-insoluble glucan to prolong biofilm adhesion on the tooth surface (*Bowen & Koo, 2011*). Moreover, the genes encoding glucan-binding proteins (*gbps*; *gbpA*, *gbpB*, and *gbpC*) (*Banas & Vickerman, 2003*) and bacterial surface antigen (*spaP*) participate in biofilm attachment (*Yang et al., 2019*). Quorum sensing, cellular metabolism, DNA repair, and stress tolerance in *S. mutans* are regulated by luxS-based signaling (*Wen et al., 2011*; *Yoshida et al., 2005*).

Natural compounds are considered as an adjunctive anti-plaque agent with good biocompatibility and less bacterial resistance (*Jeon et al., 2011*; *Utispan et al., 2017*). Natural products exhibit a wide range of biological activities, such as anti-microbial and anti-biofilm effects, which may useful for caries preventive agent development (*Jeon et al., 2011*). However, matrix production and tight adhesion of dental biofilm may reduce the effect of natural product application (*Koo et al., 2002a*). Caffeic acid phenethyl ester (CAPE), a common active compound extracted from propolis, has demonstrated antibacterial, anti-inflammatory, antioxidant and anticancer effects (*Niu et al., 2020*; *Olgierd et al., 2021*; *Yin et al., 2022*). CAPE has been proposed to inhibit bacterial activity against *Staphylococcus aureus* and *Escherichia coli* by generating reactive oxygen species that damage the outer membrane of bacteria (*Lee et al., 2013*). CAPE inhibits the growth of cariogenic bacteria in the planktonic phase; *S. mutans*, *S. sobrinus*, *Actinomyces viscosus*, and *L. acidophilus* (*Niu et al., 2020*). Moreover, CAPE inhibited the formation of *S. mutans* biofilm and its key virulence factors involved in acid production, acid tolerance and adhesion (*Niu et al., 2020*). In addition, CAPE inhibited cross-kingdom biofilm formation of *S. mutans* and *Candida albicans* and was biocompatible with a human oral keratinocyte cell line (*Yin et al., 2022*). However, the effect of CAPE on multispecies biofilm formation and cariogenicity remains unclear. Based on previous studies, we hypothesized that CAPE

might inhibit multispecies biofilm formation and cariogenicity. Thus, the aim of this study was to evaluate the effect of CAPE on the formation and cariogenicity in a multispecies biofilm containing *Streptococcus mutans*, *Streptococcus oralis*, and *Streptococcus mitis*.

## MATERIALS AND METHODS

This study was an experimental *in vitro* study. A multispecies cariogenic biofilm comprising *S. mutans*, *S. mitis* and *S. oralis* was established. The effects of CAPE on multispecies biofilm adhesion, cariogenicity, cariogenic gene expression and biofilm viability and thickness were evaluated.

### Bacterial culture

*S. mutans* (ATCC 25175), *S. oralis* (ATCC 35037) and *S. mitis* (ATCC 49456T) were used in this study. Mitis salivarius bacitracin agar (HiMedia Technology, Shenzhen, China) was used as the selective media for *S. mutans* and mitis salivarius agar (HiMedia Technology, China) served as the selective media for *S. oralis* and *S. mitis*. The bacteria were separately or co-inoculated in brain heart infusion broth (BHI, Becton, Dickinson, Franklin Lakes, NJ, USA) overnight at 37 °C under aerobic conditions with 5% $CO_2$ before performing the experiments.

### Bacterial susceptibility test

Bacteria were maintained in BHI broth overnight, and the bacterial number was assessed at an optical density (OD) of 600 nm. The bacterial suspension of *S. mutans*, *S. oralis* or *S. mitis* was adjusted to $2 \times 10^7$ CFU/ml and transferred to a 96-well plate (100 μl/well). CAPE was purchased from Sigma-Aldrich (St. Louis, MO, USA). A stock solution of CAPE was prepared in 50% dimethyl sulfoxide (DMSO; Sigma-Aldrich, St. Louis, MO, USA)-phosphate buffered saline (pH 7.4). To test bacterial susceptibility, CAPE at stock concentration was diluted in BHI media to final concentrations of 0.8, 1.6, 3.2 or 6.4 mg/ml in 5% DMSO in each well, based on a previous study (*Niu et al., 2020*). Bacteria cultured in BHI, 5% DMSO-BHI and 50 μg/ml chlorhexidine (CHX; Thermo Fisher Scientific, Waltham, MA USA) served as growth, solvent and positive controls, respectively. The plate was incubated at 37 °C for 48 h. The plate was measured for absorbance at 600 nm in a MULTISKAN Sky microplate reader (Thermo Fisher Scientific, Waltham, MA, USA). The OD 600 nm was converted to percent bacterial growth normalized to the solvent control. To determine the concentration of CAPE needed to inhibit bacterial growth by 50%, the 50% inhibitory concentration ($IC_{50}$) of CAPE against the bacteria was analyzed. The $IC_{50}$ was calculated from the concentration-response curve. Three independent experiments were performed.

### Multispecies biofilm formation study

Dental biofilm forms on the hydroxyapatite (HA)-containing enamel surface. To mimic the dental surface for biofilm adhesion, HA-coated plates were prepared by modifying the protocol in a previous study (*Schilling et al., 1994*). Briefly, a calcifying solution containing 2.5 mM $CaCl_2 \cdot 2H_2O$, 7.5 mM $KH_2PO_4$ and 250.0 mM triethanolamine at pH 7.4 was prepared and placed in 96-well microplates (Corning, Oneonta, NY, USA). The plates were

incubated at 75 °C for 3.5 h for two cycles. The plates were dried at room temperature and sterilized with ethylene oxide. Bovine serum albumin (1%) in phosphate buffer was added in the HA-coated microplates and incubated for 1 h at 37 °C.

The multispecies bacteria comprising *S. mitis*, *S. oralis* and *S. mutans* was mixed at a 1:1:1 ratio in BHI broth supplemented with 1% sucrose and placed in the HA-coated wells. CAPE dilutions were added to each well at final concentrations of 0.8, 1.6 and 3.2 mg/ml. Bacteria grown in BHI, 5% DMSO-BHI and 50 μg/ml CHX served as growth, solvent and positive controls, respectively. The plate was incubated at 37 °C for 48 h. Non-adherent bacteria were washed out with phosphate buffered saline. Multispecies biofilm adhered on HA-coated wells was fixed with absolute methanol for 15 min and stained with 0.1% crystal violet for 5 min. The stained biofilm was solubilized with 33% glacial acetic acid. The OD at 592 nm was determined using a microplate reader. Biofilm formation was reported as a percentage normalized to the solvent control. Three independent experiments were conducted.

## Total RNA extraction and cDNA synthesis

*S. mitis*, *S. oralis* and *S. mutans* co-culture was performed at a 1:1:1 ratio in BHI broth supplemented with 1% sucrose. The bacterial co-culture was treated with 1.6 mg/ml CAPE at 37 °C for 24 h. Bacteria treated with 5% DMSO served as the solvent control. Total RNA was extracted using Trizol reagent (Invitrogen, Carlsbad, CA, USA). Total RNA concentration and purity were determined by a NanoDrop 2000 (Thermo Fisher, Waltham, MA, USA) with a 260/280 ratio in a range of 1.8–1.9. Total RNA (500 ng) was used to synthesize cDNA using a PrimeScript 1st strand cDNA Synthesis Kit (Takara Bio Inc., Shiga, Japan). The total volume of the reverse transcription (RT) mixture was 10 μl: total RNA, 6.5 μl; 5× primer Script Buffer, 2 μl; PrimeScript RT Enzyme Mix, 0.5 μl; Oligo dT Primer, 0.5 μl; and Random 6 mers, 0.5 μl. The RT conditions were: RT at 37 °C for 15 min, inactivation of reverse transcriptase at 85 °C for 5 s and hold at 4 °C. The cDNA was stored at −20 °C until use.

## Real-time quantitative PCR (qPCR)

The PCR primers used in this study were from a previous study (*Wen et al., 2010*) (Table 1). *16s rRNA* gene was used as an internal control (*Thanetchaloempong, Koontongkaew & Utispan, 2022*). To test the specificity of the primers, SYBR green based real time PCR was performed in the QuantStudio™ 3 Real-Time PCR System (Thermo Fisher Scientific, Waltham, MA USA). The specific real time PCR condition was selected when the specific melting curve of the PCR product resulted in the tested primer group, while a disappearing melting curve/ cycle threshold (CT) was detected in the non-template control group.

Real-time qPCR was performed using a KAPA SYBR® FAST qPCR Kit Master Mix (Kapa Biosystems, Wilmington, MA, USA). The total volume of the qPCR system was 20 μl: cDNA (10 ng/μl), 2 μl; sense/antisense primer (10 μM each), 0.4 μl each; SYBR Green Master Mix (Kapa Biosystems, Wilmington, MA, USA), 10 μl; and ddH$_2$O, 7.2 μl. qPCR was performed in a two-step method in the QuantStudio™ 3 Real-Time PCR System

**Table 1 Primer sequences used for real-time PCR.**

| Primers | Forward primers (5′-3′) | Reverse primers (5′-3′) | Size (bp) |
|---|---|---|---|
| gtfB | AGCAATGCAGCCATCTACAAAT | ACGAACTTTGCCGTTATTGTCA | 98 |
| ldh | TTGGCGACGCTCTTGATCTTAG | GTCAGCATCCGCACAGTCTTC | 92 |
| brpA | CGTGAGGTCATCAGCAAGGTC | CGCTGTACCCCAAAAGTTTAGG | 148 |
| spaP | TCCGCTTATACAGGTCAAGTTG | GAGAAGCTACTGATAGAAGGGC | 121 |
| luxS | ACTGTTCCCCTTTTGGCTGTC | AACTTGCTTTGATGACTGTGGC | 93 |
| gbpB | CGTGTTTCGGCTATTCGTGAAG | TGCTGCTTGATTTTCTTGTTGC | 108 |
| 16S rRNA | TCCACGCCGTAAACGATGA | TTGTGCGGCCCCCGT | 119 |

(Thermo Fisher Scientific, Waltham, MA USA). The qPCR conditions were: initial denaturation at 95 °C for 10 min, followed by 40 cycles of denaturation at 95 °C for 15 s and annealing at 60 °C for 60 s. The melting curve analysis was performed at the end of the amplification from 60 °C to 95 °C and with a hold of 1 s every 0.16 °C. The original CT values of all genes were obtained from QuantStudio™ Design & Analysis Software (Thermo Fisher Scientific, Waltham, MA USA). The relative gene expression was evaluated by the difference in cycle threshold (ΔCT) between the CAPE-treated multispecies bacteria and the solvent control using the $2^{-\Delta\Delta CT}$ equation [ΔCT = CT (CAPE treatment) − CT (solvent control)] (*Rao et al., 2013*). Three independent experiments were conducted.

## Glucosyltransferase activity assay

Multispecies or monospecies bacteria-derived glucosyltransferase (GTF) was prepared by modifying the protocol from a previous study (*Linzer & Slade, 1976*). Briefly, single bacteria ($1 \times 10^8$ CFU/ml) or mixed *S. mitis*, *S. oralis* and *S. mutans* (1:1:1 ratio) were cultured in sucrose-free tryptic soy broth at 37 °C for 48 h. The cultured suspension was centrifuged at 10,000 × g for 30 min, and the supernatant was collected. Crude GTF was precipitated with 45% ammonium sulphate at 4 °C for 48 h. The precipitated enzyme was dissolved in 15 ml phosphate buffer, pH 7.4 and dialyzed for 4 days at 4 °C against 0.05 M phosphate buffer, pH 6.8 (4 L). The buffer was changed every 24 h. The dialyzed enzyme was lyophilized for 48 h. The enzyme powders were dissolved in 0.05 M phosphate buffer, pH 7.4. The protein concentration was determined using a Pierce™ BCA protein assay kit (Thermo Fisher Scientific, Waltham, MA USA) according to the manufacturer's directions. The enzyme was kept at 80 °C and diluted in phosphate buffer to 200 mg/ml prior to use.

To determine GTF activity, 50 μl of the enzyme (200 mg/ml), was mixed with CAPE at final concentrations of 0.8, 1.6 and 3.2 mg/ml in sucrose (30 μmole) containing 0.6 M acetate buffer pH 5.5 to a final volume of 300 μl. The buffer without CAPE was used as control. The reaction mixture was incubated at 37 °C for 3 h and further heated at 100 °C for 5 min. The enzyme was heated to 100 °C for 5 min. The glucans produced from GTF activity were precipitated by centrifugation at 20,000 × g for 6 min. The glucans were collected by treating them with a 5% phenol-containing sulphuric acid solution, heated to

110 °C for 5 min and cooled to room temperature. Glucan production was measured *via* the OD at 490 nm in a microplate reader (Thermo Fisher Scientific, Waltham, MA, USA). The GTF activity was expressed as the percentage of glucan production compared with the control group using the formula [(OD treatment − OD blank)/(OD control − OD blank)] × 100. Three independent experiments were conducted.

## Confocal laser scanning microscopy

HA-coated surfaces were formed in eight-well chamber glass slides (SPL Lifesciences Co. Ltd, Gyeonggi-do, Korea) as previously described. The multispecies bacteria comprising *S. mitis*, *S. oralis* and *S. mutans* (1:1:1 ratio) were co-cultured in BHI supplemented with 1% sucrose in the HA-coated slides and incubated at 37 °C for 24 h. Non-adherent bacteria were removed from the well and washed with phosphate buffer. CAPE (0.8 and 1.6 mg/ml) was added in each well and incubated at 37 °C for 24 h. The biofilm in 5% DMSO-BHI and 50 μg/ml CHX served as solvent and positive controls, respectively. The biofilm was stained using the Live/Dead BacLight Bacterial Viability kit (Invitrogen Molecular Probes, Carlsbad, CA, USA). Live and dead bacteria were visualized as green and red signals, respectively, using a confocal laser scanning microscope (CLSM) (FLUOVIEW FV3000 Olympus, Tokyo, Japan). To analyze the bacterial viability within the biofilm and total biofilm thickness, the Z-stack images of the stained biofilm were obtained using the cellSens Dimension software (Olympus, Tokyo, Japan). Three independent experiments were conducted.

## Statistical analysis

The data are quantitative and presented as the means and standard error of the mean (SEM). The unpaired t-test was performed to compare the gene expression level between the solvent control and the CAPE-treated group. For multiple group comparison of percent bacterial growth, percent GTF activity, live/dead bacterial ratio and biofilm thickness, one-way ANOVA was applied, followed by Dunnett's *post hoc* test using Prism GraphPad 8.0 (GraphPad Software, La Jolla, CA, USA). Significance was determined at $p \leq 0.05$.

# RESULTS

## Antibacterial effect of CAPE

To confirm the antibacterial effect of CAPE, the cytotoxic effect of the CAPE solvent (5% DMSO) was evaluated on *S. mitis*, *S. oralis* and *S. mutans*. The solvent control maintained bacterial growth similar to the control group ($p > 0.05$), thus, it was non-toxic to the tested bacteria (Fig. 1). Therefore, the solvent control was used to compare the CAPE effects in subsequent experiments. CAPE at 0.8, 1.6, 3.2 and 6.4 mg/ml exhibited a significant antibacterial effect on *S. mitis* (Fig. 1A) and *S. mutans* (Fig. 1C) compared with the solvent control ($p < 0.05$). CAPE at 1.6, 3.2 and 6.4 mg/ml significantly decreased the percent bacterial growth in *S. oralis* compared with the solvent control ($p < 0.05$) (Fig. 1B). The 50% inhibitory concentration ($IC_{50}$) value of CAPE was also calculated. The $IC_{50}$ of CAPE (mean ± SEM) against *S. mitis*, *S. oralis* and *S. mutans* was 0.87 ± 0.02, 1.39 ± 0.09 and

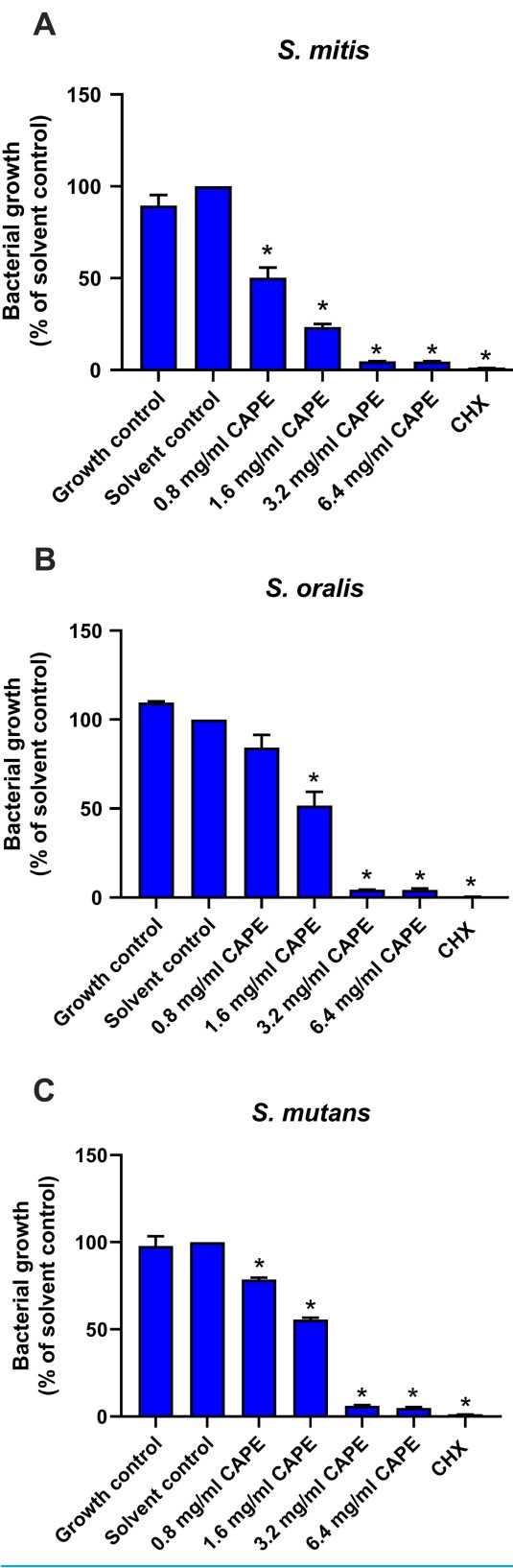

**Figure 1 Antibacterial effect of CAPE.** Effect of CAPE (0.8–6.4 mg/ml) on planktonic growth of (A) *S. mitis*, (B) *S. oralis* and (C) *S. mutans*. Bacteria cultured in media without CAPE was used as growth

**Figure 1** (continued)
control. Bacteria cultured in media with 5% DMSO was used as solvent control. Chlorhexidine (CHX) at
50 µg/ml was used as positive control. Bars represent the mean ± SEM of the percent bacterial growth
($n = 3$). *$p < 0.05$ compared with solvent control. 

$1.66 \pm 0.14$ mg/ml, respectively. Based on the $IC_{50}$ data, *S. mutans* had the highest resistance to CAPE among the tested bacteria.

## CAPE affected adhesion and cariogenic gene expression in the multispecies biofilm

A multispecies biofilm comprised of *S. mitis*, *S. oralis* and *S. mutans* (1:1:1) was established. The effect of CAPE on the adhesion and gene expression of virulence factors in the multispecies biofilm were investigated. The results demonstrated that 0.8, 1.6 and 3.2 mg/ml CAPE significantly reduced multispecies biofilm formation by 0.5–70% compared with the solvent control ($p < 0.05$) (Fig. 2A). Moreover, 1.6 mg/ml CAPE significantly decreased gene expression involved in sucrose-dependent adhesion (*gtfB* and *gbpB*), acidogenicity (*ldh*), aciduricity (*brpA*) and quorum sensing (*luxS*) in multispecies biofilm compared with the solvent control ($p < 0.05$) (Fig. 2B). However, there was no significant difference in sucrose-independent adhesion (*spaP*) gene expression in the multispecies biofilm between the CAPE treatment and solvent control group ($p > 0.05$).

## CAPE reduced GTF activity in monospecies and multispecies bacterial cultures

The intrinsic GTF activity in monospecies and multispecies bacterial cultures is presented in Table 2. *S. oralis* and *S. mutans* expressed relatively high GTF activity compared with that of *S. mitis*. Moreover, the maximum GTF activity was found in the multispecies bacterial culture model. Therefore, the changes in GTF activity after CAPE treatment in the monoculture of *S. oralis* and *S. mutans* and multisspecies co-culture comprising *S. mitis*, *S. oralis* and *S. mutans* were evaluated. The results indicated that 0.8 and 1.6 mg/ml CAPE significantly decreased GTF activity in *S. oralis* compared with control ($p < 0.05$) (Fig. 3A). However, 3.2 mg/ml CAPE did not significantly change GTF activity in *S. oralis*. Interestingly, CAPE at all tested concentrations significantly inhibited GTF activity in *S. mutans* to approximately 48% compared with control ($p < 0.05$) (Fig. 3B). In addition, CAPE significantly reduced GTF activity in the multispecies co-culture system in a dose-dependent manner compared with control ($p < 0.05$) (Fig. 3C).

## Bactericidal penetrative effect of CAPE in multispecies biofilm

The bactericidal penetrative effect of 0.8 and 1.6 mg/ml CAPE was assessed in the multispecies biofilm using CLSM. Z-stack images displayed the bactericidal depth of CAPE from the outermost into innermost layers of the multispecies biofilm (Fig. 4A). We found that 0.8 and 1.6 mg/ml CAPE significantly decreased the live/dead bacterial ratio in the multispecies biofilm compared with the solvent control ($p < 0.05$) (Fig. 4B).

**A**

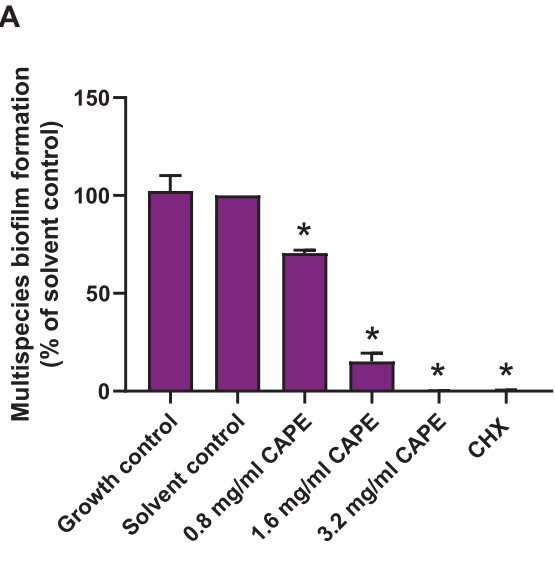

**B**

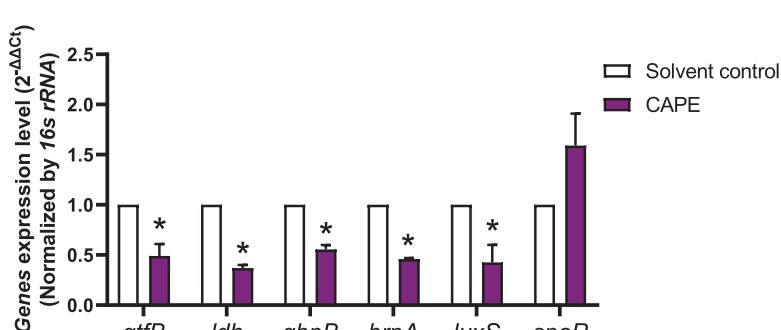

**Figure 2 Effect of CAPE on multispecies biofilm formation and cariogenic gene expression.** CAPE (0.8–3.2 mg/ml) was used to treat the multispecies biofilm comprising *S. mitis*, *S. oralis* and *S. mutans* (1:1:1) for 48 h. Multispecies biofilm in media without CAPE was used as growth control. Multispecies biofilm in media with 5% DMSO was used as solvent control. Chlorhexidine (CHX) at 50 µg/ml was used as positive control. (A) Multispecies biofilm adhesion was evaluated using crystal violet staining. The multispecies bacteria were treated with 1.6 mg/ml CAPE for 24 h. (B) Cariogenic gene expression in the multispecies biofilm was evaluated using real time PCR. Bars represent the mean ± SEM of the percent multispecies biofilm formation or gene expression level ($n = 3$). *$p < 0.05$ compared with the solvent control.

**Table 2 Glucosyltransferase activity in monospecies and multispecies bacterial cultures.**

| Bacteria | Baseline GTF activity measured at OD 490 nm (mean ± SEM) |
|---|---|
| *S. mitis* | 0.03 ± 0.03 |
| *S. oralis* | 1.44 ± 0.17 |
| *S. mutans* | 0.70 ± 0.08 |
| Multispecies | 1.67 ± 0.08 |

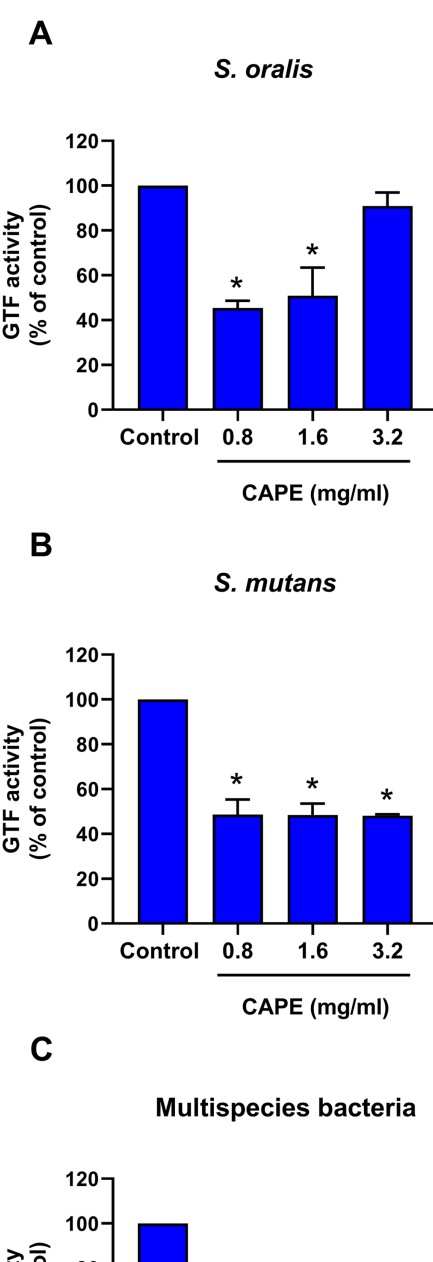

**Figure 3 Effect of CAPE on GTF activity.** Single species or multispecies bacteria (*S. mitis*, *S. oralis* and *S. mutans*) was cultured and the GTF enzyme was collected. CAPE (0.8–3.2 mg/ml) was added in the enzyme reaction mixture. GTF reaction without CAPE was used as control. GTF activity was determined in monoculture of (A) *S. oralis* and (B) *S. mutans* and (C) co-culture of multispecies bacteria. Bars represent the mean ± SEM of percent GTF activity (*n* = 3). *$p < 0.05$ compared with control.

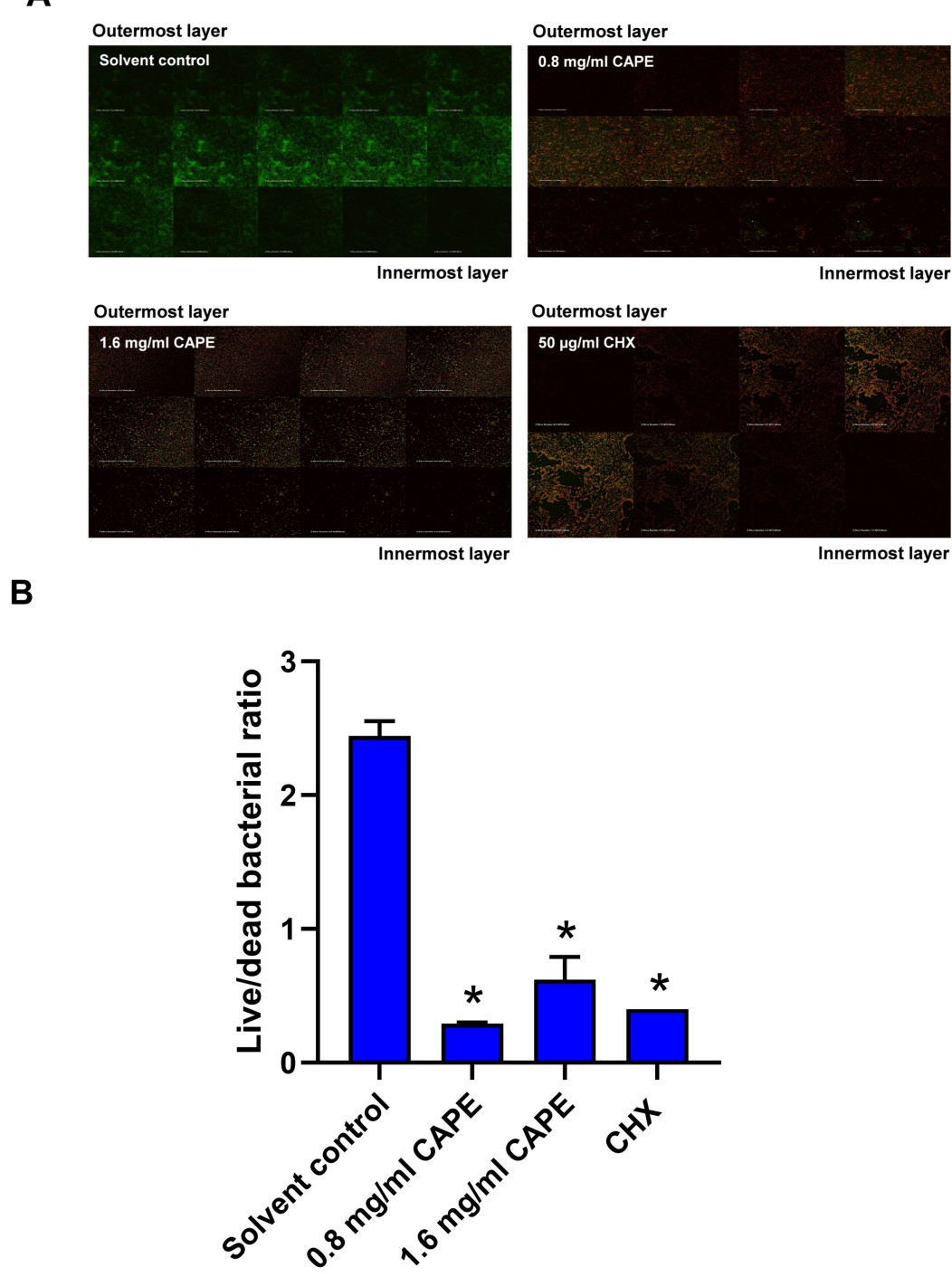

**Figure 4 Bactericidal penetrative effect of CAPE on multispecies biofilm.** CAPE (0.8 and 1.6 mg/ml) was used to treat the multispecies biofilm that comprised *S. mitis*, *S. oralis* and *S. mutans* (1:1:1). Multispecies biofilm in media with 5% DMSO was used as solvent control. Chlorhexidine (CHX) at 50 µg/ml served as positive control. (A) Confocal laser scanning microscopy displayed live (green) and dead (red) bacteria in each layer of multispecies biofilm. (B) Ratio of live/dead bacteria in the CAPE-treated multispecies biofilm was calculated and compared with solvent control. Bars represent the mean ± SEM of live/dead bacterial ratio in multispecies biofilm ($n = 3$). *$p < 0.05$ compared with solvent control.

**A**

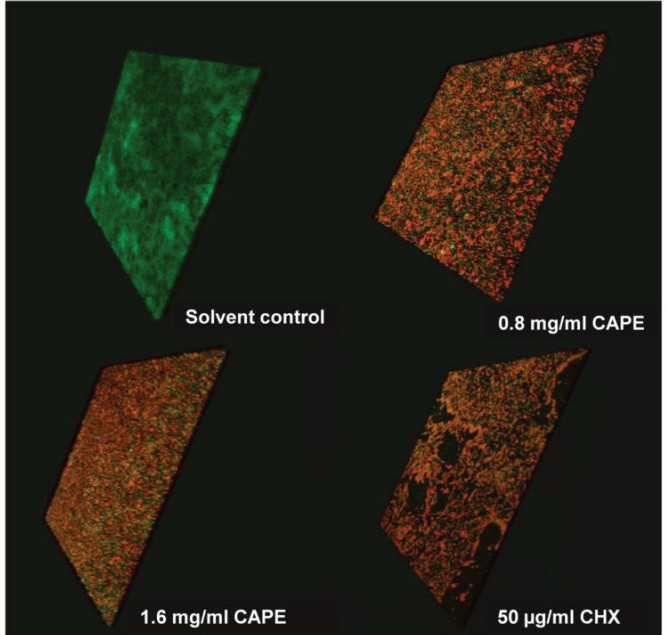

**B**

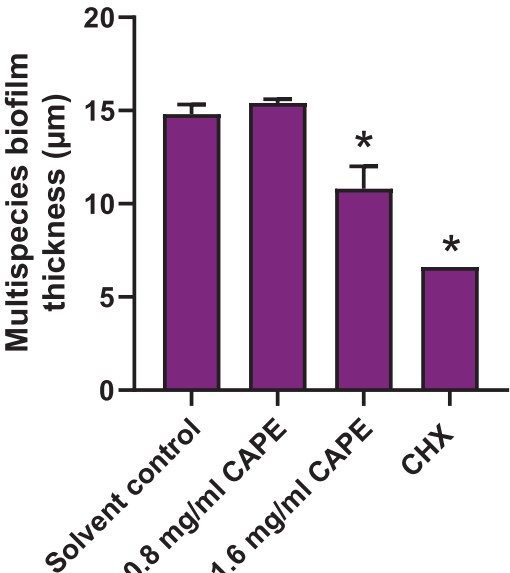

**Figure 5 Effect of CAPE on the reduction of multispecies biofilm thickness.** CAPE (0.8 and 1.6 mg/ml) was used to treat multispecies biofilm that comprised *S. mitis*, *S. oralis* and *S. mutans* (1:1:1). Multispecies biofilm in media with 5% DMSO was used as solvent control. Chlorhexidine (CHX) at 50 μg/ml served as positive control. (A) Confocal 3D imaging displayed structures with live (green)/dead (red) bacteria in multispecies biofilm. (B) The multispecies biofilm thickness was calculated in micrometers (μm) and compared with solvent control. Bars represent the mean ± SEM of biofilm thickness (*n* = 3). *$p < 0.05$ compared with solvent control.

**Effect of CAPE on the reduction of the multispecies biofilm thickness**

The thickness of the 0.8 and 1.6 mg/ml CAPE-treated multispecies biofilm was evaluated using CLSM. The confocal images exhibited 3D structures with live/dead bacteria in the multispecies biofilm at all conditions (Fig. 5A). Measurement of the multispecies biofilm structure indicated that 1.6 mg/ml CAPE significantly reduced the biofilm thickness compared with the solvent control ($p < 0.05$) (Fig. 5B).

## DISCUSSION

CAPE is a common bioactive compound detected in propolis extract and has anti-cariogenic effects on oral bacteria and *S. mutans* biofilm (*Niu et al., 2020*; *Yin et al., 2022*). The present study created a multispecies cariogenic biofilm model composed of *S. mitis*, *S. oralis* and *S. mutans*. The results demonstrated that CAPE inhibited the multispecies biofilm formation, reduced cariogenicity, cariogenic gene expression and biofilm viability and thickness. In the planktonic bacterial susceptibility assay, CAPE exhibited anti-bacterial activity against *S. mutans* at $IC_{50}$ 1.66 ± 0.14 mg/ml, which is consistent with previous studies (*Niu et al., 2020*; *Yin et al., 2022*). Moreover, the present study is the first to report the anti-bacterial property of CAPE against *S. mitis* and *S. oralis*. Based on their $IC_{50}$ values, *S. mitis* and *S. oralis* had higher susceptibility to CAPE compared with that of *S. mutans*. However, there was no direct evidence that explained the anti-microbial effect of CAPE on *S. mitis*, *S. oralis* and *S. mutans*. The anti-microbial property of propolis was studied in a wide range of oral bacteria (*Navarro-Perez et al., 2021*) and might be indirectly used to compare with our data. This previous evidence indicated that propolis extracts had a lower bactericidal effect on *S. mutans* compared with other oral bacteria. The present study might confirm the high CAPE resistance in *S. mutans* found in the prior study.

The multispecies biofilms have an effective response to environmental change. This synergistic function of multispecies biofilm improves its ability to respond to that change compared with that of monospecies by effectively reducing the metabolic products in the biofilm (*Ouidir, Gabriel & Nait Chabane, 2022*). Cariogenic biofilm containing *S. mutans* (*Niu et al., 2020*) or cross-kingdom microbes (*S. mutans* and *C. albicans*) (*Yin et al., 2022*) were used as the model to demonstrate the CAPE effect on the monospecies and dualspecies biofilm, respectively. They found that 0.04 mg/ml and 20 µg/ml CAPE significantly inhibited monospecies and cross-kingdom, respectively, biofilm formation. Using the multispecies cariogenic biofilm model, we found that 0.8 mg/ml CAPE significantly decreased biofilm formation. Our data imply that the multispecies biofilm exhibited higher CAPE resistance than that of single or cross-kingdom biofilm.

Cariogenic bacteria, such as *S. mutans*, lactobacilli or low pH non-mutans streptococci, synergistically respond to environmental changes in the biofilm (*Marsh, 1991*; *Takahashi & Yamada, 1999*). To survive in the stress environment biofilm, the cariogenic bacteria express acidogenicity (*Duguid, 1985*), aciduricity (*Lemos & Burne, 2008*), adhesion (*Banas & Vickerman, 2003*; *Bowen & Koo, 2011*; *Yang et al., 2019*) and adaptation (*Wen et al., 2011*; *Yoshida et al., 2005*) through the effective function of several genes and corresponding proteins. This study demonstrated that CAPE attenuated the

cariogenic-associated gene expression levels of *ldh* (acidogenicity), *brpA* (aciduricity), *gtfB* and *gbpB* (sucrose-dependent adhesion) and *luxS* (quorum sensing-related signaling pathway) in the multispecies biofilm. However, CAPE did not affect the gene expression of *spaP*. In absence of sucrose, *S. mutans* directly binds to the tooth surface using *spaP* (*Bowen et al., 1991*). When sucrose is present, GTF, a key enzyme found in *S. mutans* and *S. oralis*, catalyzes sucrose into glucans (*Bowen & Koo, 2011*; *Fujiwara et al., 2000*). Glucans bind to glucan-binding proteins to create bacterial co-aggregation within the biofilm (*Banas & Vickerman, 2003*). In this study, 1% sucrose was supplemented in the culture media, therefore sucrose-associated adhesion molecules might be the target of CAPE. Our findings indicate the anti-cariogenic potential of CAPE on acidogenicity, acid tolerance, sucrose-dependent adhesion and quorum sensing mechanism in the multispecies biofilm.

Water-insoluble glucans are the most important molecules for biofilm adhesion and are targeted by several anti-cariogenic natural compounds (*Koo et al., 2002b*). In the present study, intrinsic GTF activity was detected in *S. mutans* and *S. oralis* and multispecies bacteria, but not in *S. mitis*. Low CAPE concentration significantly inhibited GTF function in the single and multispecies bacteria. Moreover, high CAPE concentration specifically decreased GTF activity in *S. mutans* and multispecies bacteria. The competitive growth of cariogenic bacteria is a critical phenomenon in a multispecies biofilm (*Jakubovics, 2015*). *S. mutans* exhibited bactericidal factors or modified its essential mechanisms to overcome the growth of other bacteria in a mixed species biofilm (*Choi et al., 2023*; *Kreth et al., 2005*). From the previous evidence, we hypothesize that *S. mutans* might be a major target of action of CAPE for inhibiting GTF in the multispecies biofilm. In mature biofilm formation, bacterial colonization and extracellular polysaccharide (EPS) are completely organized (*Bowen & Koo, 2011*). Complete EPS synthesis determines the protective structure and environmental niche in dental biofilm (*Vu et al., 2009*). EPS production is performed by GTF in a wide range of cariogenic bacteria (*Bowen & Koo, 2011*). This study demonstrated that CAPE inhibited GTF function, penetrated mature biofilm and significantly reduced live bacteria and biofilm thickness. These results suggest that decreased multispecies biofilm thickness and viable bacteria might be a consequence of CAPE-suppressed GTF activity.

## CONCLUSIONS

The present study demonstrated that CAPE had anti-cariogenic effects on the multispecies biofilm formation. We found that CAPE suppressed cariogenic gene expression involved in acidogenicity, acid tolerance, sucrose-dependent adhesion and *luxS*-derived quorum sensing. Moreover, CAPE inhibited GTF functions and exerted penetrative bactericidal and biofilm thickness reduction. Our findings suggest that CAPE might be applied as an adjunctive agent in dental products for caries prevention. However, the role of CAPE in suppressing the specific bacteria and components in cariogenic multispecies biofilm requires further investigation.

## ACKNOWLEDGEMENTS

We thank Dr. Suppanut Jongjitaree, International College of Dentistry, Walailak University for preliminary information on CAPE.

### Funding

This work was supported by the Faculty of Dentistry Thammasat University Research Fund (No. 2/2566). The funders had no role in study design, data collection and analysis, decision to publish, or preparation of the manuscript.

### Grant Disclosures

The following grant information was disclosed by the authors:
Faculty of Dentistry Thammasat University Research Fund: 2/2566.

### Competing Interests

The authors declare that they have no competing interests.

### Author Contributions

- Paopanga Kokilakanit performed the experiments, authored or reviewed drafts of the article, and approved the final draft.
- Nonthakorn Dungkhuntod performed the experiments, prepared figures and/or tables, and approved the final draft.
- Nitchadakorn Serikul performed the experiments, prepared figures and/or tables, and approved the final draft.
- Sittichai Koontongkaew analyzed the data, authored or reviewed drafts of the article, and approved the final draft.
- Kusumawadee Utispan conceived and designed the experiments, analyzed the data, prepared figures and/or tables, authored or reviewed drafts of the article, and approved the final draft.

### Data Availability

The raw measurements and statistical analysis of all experiments are available in the Supplemental Files.

### Supplemental Information

Supplemental information for this article can be found online at http://dx.doi.org/10.7717/peerj.18942#supplemental-information.

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
