# Peer review of "Caffeic acid phenethyl ester inhibits multispecies biofilm formation and cariogenicity"

_PeerJ, doi:10.7717/peerj.18942_

## Round 0.1 · original submission · Major Revisions

Dear authors,

The manuscript presents an original and relevant study on the antimicrobial potential of CAPE, but major revisions are required to address significant gaps. The introduction needs clearer context, including null hypotheses and the rationale for CAPE’s selection. The materials and methods section lacks key details on study design, CAPE preparation, and statistical analysis, making the methodology unclear. Results should incorporate statistical tables and explicitly describe the significance of findings, including controls. The discussion should better connect findings to broader implications and similar studies, while the conclusion should succinctly summarize contributions. Additionally, English language corrections are essential for clarity.

Reviewer 1 ·

Basic reporting

The introduction provides a good overview of the context but could benefit from additional clarity. Consider revising phrases for better readability (e.g., "homeostatic ecology in normal biofilm changes when exposed to sugar").
Clarify why only certain species, such as S. mutans and Lactobacillus spp., are highlighted for their cariogenic potential.
Ensure terms like "virulence factors" are consistently defined throughout for clarity.
Specify image analysis software used for quantifying biofilm thickness in the CLSM subsection.
Ensure consistent mention of significance levels in the "Statistical analysis" subsection.
Clarify IC50 value explanations and differences for S. mitis, S. oralis, and S. mutans in the results section.

Experimental design

Clarify incubation conditions for bacterial culture (e.g., temperature and CO2 settings) and whether they are consistent across species.
Provide justification for selecting specific CAPE concentrations in the "Bacterial susceptibility test" and reference prior studies or preliminary data.
Add explanation for why hydroxyapatite-coated plates were used in the "Multispecies biofilm formation study" to mimic dental conditions.
Confirm that the 16s rRNA gene is the most suitable internal control for co-cultures of S. mutans, S. oralis, and S. mitis.
Include validation for optimized enzyme concentration and incubation conditions in the "Glucosyltransferase activity assay."

Validity of the findings

Include a rationale for CAPE’s varying antibacterial effects on different bacterial species (e.g., cell wall structures, metabolic activities).
Explain CAPE's differential impact on biofilm adhesion and gene expression in the multispecies biofilm.
Discuss possible reasons why CAPE did not significantly affect the expression of the spaP gene.
Highlight hypotheses for why S. mutans might be the major target of CAPE in the multispecies biofilm for GTF inhibition.
Ensure quantitative data or statistical significance accompanies the presentation of Z-stack images and depth analysis.

Additional comments

Add a sentence to the introduction that connects biofilm formation challenges to the potential benefits of natural compounds.
Expand on CAPE's use in other antimicrobial applications beyond oral health.
Reiterate the study’s novelty in evaluating CAPE's impact on multispecies biofilms compared to single-species models.
Mention limitations or challenges related to CAPE’s bioavailability or application in dental products in the conclusion.
Provide context for the competitive growth behavior of S. mutans and its impact on CAPE's long-term efficacy in preventing biofilm formation.
Recommend citing “DOI: 10.1016/j.jval.2020.08.689” for insights into clinical applicability and antimicrobial evaluations across settings.

Reviewer 2 ·

Basic reporting

The English language used requires comprehensive corrections by a native English speaker.
The references and literature cited are appropriate.
There is a lack of statistical tables mentioned in the study (a separate statistical table should be included for each variable).
The hypotheses were not stated at the end of the introduction, nor were they discussed in the discussion section.

Experimental design

The submitted topic is original and of good quality.
The research question is clear within the context but should be explicitly stated at the end of the introduction alongside the study hypothesis.
The manuscript does not mention any approval for this study from a scientific committee or educational institution.
The study design is unclear and should be outlined at the beginning of the materials and methods section.
Simplifying the scientific writing overall is recommended to ensure that non-specialist readers can understand the article's content and methodology.

Validity of the findings

The presented results are valuable and contribute to the medical field. However, I have some comments in this regard: the materials and methods section mentions a positive control group with chlorhexidine (concentration unspecified), yet the results do not provide any information about it. Additionally, the distribution of plates across each test was not clarified in the materials and methods section. Therefore, including statistical tables in the results section would provide greater clarity on several ambiguous aspects of this article.
The conclusion section should better reflect a summary of the findings from this study than what is currently presented.

Additional comments

The importance of studying the effect of this substance on the entire biofilm, rather than individual bacteria alone, should be clarified to emphasize the significance of the study.
The discussion section should cover all the observed results in this study, along with findings from similar studies.

Reviewer 3 ·

Basic reporting

Introduction:
- The introduction requires improvement, particularly in the conceptualization of dental caries. I recommend that the authors incorporate recent literature and revise the first paragraph accordingly. The consensus developed by the ORCA (European Organisation for Caries Research) could serve as a valuable resource for clarifying caries definitions. Additionally, there is a misunderstanding regarding the distinction between demineralization and the occurrence of caries. It is also crucial to differentiate between caries as a disease and caries lesions.
- Provide the null hypotheses of the study.
- It is not clear the importance of the present study if the authors already mentioned that CAPE has exhibited antimicrobial activity against cariogenic bacteria. Additionally, the rationale for selecting CAPE as the focus of this investigation should be better described. The authors should provide a more detailed explanation of why CAPE was chosen and its potential as a promising agent against dental caries.

Material and Methods:
- There is no detail about how CAPE was prepared. In which concentration was it used? Did the authors characterize the compound? What was CAPE’s pH? This information should be described in a separate item. Only in the results I could realize that CAPE was used at 0.8, 1.6, 3.2, and 6.4 mg/ml. So, are there 4 groups in the study?
- The methodology is overall sound.
- It is not possible to understand the statistical analysis as the authors did not define the study design previously. Why was the Unpaired T test used? Which were the two groups analyzed? For which dependent variables the one-way ANOVA was applied? Please specify it.

Experimental design

Please see my comments in "basic reporting".

Validity of the findings

Please see my comments in "basic reporting".

---

## Round 0.2 · accepted · Accept

Dear authors,
Congratulations on your efforts in addressing the reviewers' comments and enhancing the quality of the manuscript. The revisions have significantly improved the clarity and rigor of the study, and the research presents valuable insights into the inhibitory effects of caffeic acid phenethyl ester on biofilms and its potential as a therapeutic agent against cariogenicity. Given the comprehensive responses, improved experimental design, and the validity of the findings, we are pleased to inform you that the manuscript has been accepted for publication.

Reviewer 3 ·

Basic reporting

The authors properly answered the questions, and the manuscript report was improved.

Experimental design

The authors properly answered the questions, and the manuscript report was improved.

Validity of the findings

The authors properly answered the questions, and the manuscript report was improved.

Additional comments

The authors properly answered the questions, and the manuscript report was improved.